# Sensitization of Guinea Pig Skin to Imported Fire Ant Alkaloids and Establishment of an Inflammatory Model

**DOI:** 10.3390/ijerph20031904

**Published:** 2023-01-20

**Authors:** Yueze Liu, Jun Huang, Juan Zhang, Yipeng Xu, Xiaowei Li, Yaobin Lu

**Affiliations:** 1State Key Laboratory for Managing Biotic and Chemical Threats to the Quality and Safety of Agro-Products, Key Laboratory of Biotechnology in Plant Protection of MOA of China and Zhejiang Province, Institute of Plant Protection and Microbiology, Zhejiang Academy of Agricultural Sciences, Hangzhou 310021, China; 2College of Life Sciences, China Jiliang University, Hangzhou 310018, China; 3Institute of Garden Plants and Flowers, Zhejiang Academy of Agricultural Sciences, Hangzhou 311202, China

**Keywords:** imported fire ant, alkaloid, inflammation, skin sensitivity test, histological change

## Abstract

Imported fire ants (IFAs), *Solenopsis invicta*, release their venom through multiple stings that induce inflammation, allergies, shock, and even death. Although IFA venom protein sensitization and related subcutaneous immunotherapy have been studied, few studies have examined the potential toxicity or pathogenicity of alkaloids, the main substances in IFA venom. Here, IFA alkaloids were identified and analyzed by gas chromatography–mass spectrometry; we further determined an appropriate extraction method and its effectiveness for extracting high-purity alkaloids through comparative analysis and guinea pig skin sensitivity tests. The alkaloids released from the IFA abdomen included those present in the head and thorax, and the alkaloids in the abdomen accounted for the highest proportion of the total extract. The abdominal extirpation method yielded alkaloids with a purity above 97%, and the skin irritation response score and histopathological diagnosis suggest that intradermal injection of the extracted alkaloids produced symptoms effectively simulating those of IFA stings. The successful establishment of an inflammatory model in guinea pigs stung by IFAs provides a basis for further research on the mechanism of inflammatory diseases caused by IFAs.

## 1. Introduction

Humans are facing increasingly serious biosafety problems due to unknown germs, animal-derived viruses, and aggressive insects. For example, the red imported fire ant (IFA), *Solenopsis invicta* Buren (Hymenoptera: Formicidae), is an invasive omnivorous ant that prefers soil habitats; this species originated in South America and affects humans, wildlife, crops, and livestock [1]. *S. invicta* has attracted governmental and public attention due to its aggression toward people and the serious allergic reaction that its stings may induce. Similar to bees, *S. invicta* has a stinger at the end of the abdomen; however, *S. invicta* differs in that it can sting multiple times if left undisturbed [2,3]. People experience symptoms of pain, itching, and swelling after being stung by *S. invicta*; in severe cases, individuals experience fever, urticaria, and shock, and may even die [4,5,6]. An investigation revealed that more than 30% of people in four provinces of China suffered from *S. invicta* stings, and 10% experienced fever, with a few individuals experiencing systemic allergic reactions [7].

The release of venom by *S. invicta* stingers is the key to adverse reactions in the human body [2,8,9,10,11]. Approximately 95% of *S. invicta* venom contains piperidine alkaloids, which cause redness, swelling, inflammation, and local tissue necrosis in the skin [12]; the other 5% of the venom consists of water-soluble proteins and peptides that cause anaphylactic reactions [13,14]. The majority of skin reactions to *S. invicta* stings are due to the alkaloids in the venom. *S. invicta* alkaloids comprise cis- and trans-2-methyl-6-alkyl-(or alkenyl) piperidines and exhibit antibacterial, hemolytic, insecticidal, and histamine-releasing properties [9,10,15,16,17,18]. Although *S. invicta* venom proteins have been well studied [12,19,20,21], few studies have explored the sensitization or toxic effects of its alkaloids on humans. Fewer than 1% of people are allergic to venom proteins [22], and most cases are due to the inflammatory reaction caused by venom alkaloids. Additionally, it is critical to initially treat the local area of *S. invicta* stings to prevent scratching and secondary wound infection, which requires an understanding of the pathogenic mechanism of venom alkaloids.

Studies on the toxicity of *S. invicta* alkaloids need to be based on effectively extracted alkaloids. IFA venom is produced in the poison gland, stored in the venom sac, and delivered through the stinger. Several different techniques have been used for the extraction of IFA venom, such as capillary milking [15], venom gland dissection [23], and solvent soaking of whole ants [9,10]. Because whole-body extracts of *S. invicta* contain sufficient venom proteins [19], many studies have carried out subcutaneous immunotherapy with whole-body extracts of *S. invicta* to prevent the systemic reactions that occur as a result of *S. invicta* stings [20,21,24]. Although whole-body solvent soaking can adequately extract venom alkaloids from IFA workers [25], other researchers have suggested that the ratio of cis-C11 to trans-C11 in venom sampled by milking is always higher than that obtained by soaking [26]. However, to conduct animal disease model tests simulating *S. invicta* stings in the future, it is necessary to obtain high-purity alkaloids and inject them directly into animals, which may be impossible to achieve through extraction by whole-body solvent soaking. Therefore, a more practical and higher-purity method for extracting venom alkaloids from large quantities of IFAs merits further study.

We obtained *S. invicta* populations from different areas of Zhejiang Province, China, and aimed to answer the following sets of questions: (1) Are there differences in alkaloids retained in different parts of the *S. invicta* body, and what proportions of total alkaloids are present in various parts of the body? (2) How does the extraction method, i.e., abdominal extirpation, venom gland dissection, or solvent soaking of whole ants, affect the purity of alkaloids? (3) Are the alkaloids extracted by abdominal extirpation equivalent to those from *S. invicta* stings according to the animal skin sensitivity assessment and histopathological diagnosis? Through the above research, an inflammatory model in guinea pigs stung by *S. invicta* will be established. The successful establishment of an inflammatory model in guinea pigs stung by IFAs will provide a basis for further research on the mechanism of inflammatory diseases caused by IFAs.

## 2. Materials and Methods

### 2.1. Fire Ant Collection

Colonies of *S. invicta* were collected from Liandu (LD; 28.45° N, 119.85° E, altitude 60.41 m; habitat: wasteland), Longwan (LW; 28.45° N, 120.81° E, altitude 20.75 m: habitat: greensward), and Yiwu (YW; 29.24° N, 119.89° E, altitude 89.75 m: habitat: nursery) Counties in Zhejiang Province, China, in late April 2021 and transported to the laboratory in plastic containers (52.5 cm length × 32.0 cm width × 38.0 cm height) coated with Fluon^TM^ on the inside walls to keep them from escaping. A total of nine colonies were collected (three from each site). The polygyny of *S. invicta* was confirmed by the presence of multiple queens in each colony [27] and the distance (<5 m) between ant nests in a 1076 ft^2^ area. The laboratory IFA colony was established and maintained on an artificial diet following the procedures reported by Banks et al. [28]. Extractions were performed within 2 weeks after collecting ants from the field.

### 2.2. Alkaloids in Different Body Parts of S. invicta

The purpose of this experiment was to determine how alkaloids differed among body parts of *S. invicta*, including the types of alkaloids and proportions of alkaloids in the total extract. Moreover, the effect of collection location on alkaloid composition was further analyzed. A total of 270 medium-sized *S. invicta* workers (ca. 0.16 g in weight) were randomly selected from each colony and kept in different-sized batches at −20 °C. After 50 s, the frozen and remaining workers were removed from the plastic bottle and dissected under a microscope. First, we fixed the middle of the two nodules of *S. invicta* with a nipper (No. 1), quickly removed the abdomen with another nipper (No. 2), and immersed the abdomen in 15 mL of n-hexane. Then, the region from the thorax to the head was fixed with the No. 1 nipper, the thorax was removed with a new nipper (No. 3), and the head and thorax were each immersed in 15 mL of n-hexane. Filter paper was placed under the *S. invicta* body during the operation and replaced after 5 *S. invicta* were treated. Ice packs were placed around the solution to keep it cool and prevent evaporation during the extirpation process. After the ant parts were soaked for 12 h at room temperature (26 °C), the liquid was removed and filtered by a needle filter (0.22 µm). The filtrate was dried by a rotary evaporator to obtain the extract (4.2~6.4 mg). The extract was then dissolved in 0.7 mL of n-hexane for subsequent gas chromatography–mass spectrometry (GC–MS) analysis. Workers from each colony were subjected to a bioassay. This procedure was repeated three times for each site (LD, LW, and YW).

### 2.3. Comparison of Different Methods of Extracting Venom Alkaloids

Three venom alkaloid extraction methods were tested, namely, venom gland dissection, solvent soaking of whole ants, and abdominal extirpation. Venom gland dissection and solvent soaking of whole ants were conducted following the procedures reported by Liu et al. [25], which included suction filtration, secondary filtration, and rotary drying. For the whole-ant solvent-soaking method, tested workers were directly immersed in 15 mL of n-hexane at room temperature for 12 h. Workers from each colony were subjected to a bioassay. This procedure was repeated three times for each site (LD and YW).

### 2.4. Chemical Analyses

To separate and identify the *S. invicta* alkaloids, we used GC–MS (Agilent Technologies 7890B GC System coupled with an Agilent Technologies 5977B MS system) (Agilent Technologies Inc., Wilmington, NC, USA). Each injection was 1 μL, with 10:1 splitting. The sample was separated in an HP-5MS capillary column (Agilent, 30 m × 0.25 mm i.d., 0.25 μm film thickness) using helium as the carrier gas and a flow rate of 1 mL/min. The following temperature program was implemented: The initial temperature was 40 °C for 2 min; then the temperature was increased to 160 °C at a rate of 10 °C/min and held for 5 min. Next, the temperature was increased to 250 °C at a rate of 5 °C/min and held for 8 min. The temperature of the injection port and ion source was 250 °C and 230 °C, respectively. An electron impact (EI) source was utilized for ionization; the ionization energy was 70 eV, the ion source temperature was 230 °C, and the mass scanning range was 50–500 amu. Sample fragments were compared with fragment spectra in the literature [9,10] to determine the substance composition. Peak area normalization was performed for the quantification of alkaloid components, and the method and calculation formula were similar to those described by Lv et al. [29].

### 2.5. Intradermal Injection Test in Guinea Pigs

Animal experiments were further conducted not only to verify the effectiveness of the abdominal extirpation method but also to determine the toxicity of IFA alkaloids. This experiment was approved by the Ethics Committee of the Zhejiang Academy of Agricultural Sciences (ethics protocol No. 2021ZAASLA84) and performed in accordance with the principles and guidelines of the Zhejiang Farm Animal Welfare Council of China. Common-grade healthy male Hartley guinea pigs weighing between 250 and 300 g were obtained from the Animal Centre of Zhejiang Academy of Agricultural Sciences, Hangzhou, China, and allowed to acclimate for one week before experiments (animal license number: SCXK-Zhejiang-2020-0008). Guinea pigs were provided with a standard chow diet and free access to water (1 g of vitamin C per 100 mL of water) under standard laboratory conditions: Temperature, 22–25 °C; humidity, 50–60%; and light/dark cycle, 12 h/12 h. The hair on the left or right side of the dorsal portion of the guinea pigs was shaved to expose a 3 cm × 3 cm patch of skin 24 h prior to the experiment. The guinea pigs were divided into four groups of 13 guinea pigs each, as follows:SI group: Injection of normal saline (control).VI group: Vehicle control injection.AI group: Intradermal injection of alkaloids.BI group: *S. invicta* sting or bite.

Three guinea pigs in each group were only used for skin irritation score determination after treatment, while the other ten were used for histopathological diagnosis by hematoxylin and eosin (HE). The alkaloids extracted by abdominal extirpation (15.6 mg) were prepared and placed in a centrifuge tube (the purity of extracted alkaloids was verified by GC–MS and exceeded 98% before it was allowed to proceed to the next preparation), and 1560 μL of 1% hydrochloric acid and 1560 μL of 0.9% NaCl were added. Next, the fully soluble alkaloids were shaken well, and 312 μL of 1% NaOH was added to create a solution with a pH ranging from 6.8~7.0. Then, the solution was brought to 3.9 mL with 0.9% NaCl and prepared at a concentration of 4.0 mg/mL. The extracted alkaloids were injected into the exposed patch of skin on the guinea pigs at six different positions, with each injection containing 0.05 mL of the alkaloid solution (i.e., 0.20 mg of pure alkaloids), for three consecutive days (in the AI group). In the BI group, twenty *S. invicta* workers were placed on the skin patch at the same time and were allowed to sting for 15 s each day for three consecutive days. According to our preliminary data, a medium- or large-sized worker can release approximately 60 μg of alkaloids through multiple stings; thus, the amount released by 20 workers was roughly equivalent to the injection amount. Previously, we made some attempts to use one injection or small amounts of alkaloids as a standard treatment in guinea pigs. However, the persistence of the response was poor. Animals in the VI and SI groups were injected with the same amount of vehicle and normal saline, respectively. The skin patch of each guinea pig was observed for any changes, such as erythema, edema, or pustules, at 6 h postinjection and daily thereafter for 1 or 13 days. The Draize scale was applied to evaluate the degree of skin irritation [30]. Irritation scores were used to grade the stimulus intensity, which ranged from no response to a severe response, as follows: 0, no erythema or edema; 1, very slight erythema or edema; 2, well-defined erythema or mild edema (skin bulge, clear outline); 3, moderate to severe erythema or severe edema (skin bulge approximately 1 mm in size); and 4, severe erythema to slight eschar (in-depth injury) or severe edema (skin bulge larger than 1 mm, beyond the scope of the exposed area).

Histopathological changes in the skin of guinea pigs associated with IFA stings and other treatments were observed using an Eclipse Ci-L photographic microscope (Nikon, Tokyo, Japan). On the 4th and 8th days after treatment, 5 guinea pigs were randomly selected from each group for testing. The guinea pig was anesthetized by an intraperitoneal injection of pentobarbital sodium, and a skin tissue sample 1 cm × 1 cm in size was removed and fixed in 10% neutral formalin solution. After dehydration and waxing, the operation steps were embedding, slicing, dewaxing, staining with HE, and sealing with neutral glue after dehydration and clearing. Finally, the skin tissue morphology was observed.

### 2.6. Statistical Analysis

In each group, the Shapiro–Wilk test was used to determine whether the data displayed a normal distribution and homogeneous variance. In the case of normally distributed data and homogeneous variance, Tukey’s honestly significant difference (HSD) test was used to compare the effects of the different extraction methods on the purity of alkaloids; when necessary, data were normalized by either square root or logarithmic transformations. The Chi-square test was used to compare the alkaloid contents in the head, thorax, and abdomen, the Kruskal–Wallis test was used to compare the effects of AI, BI, VI, and SI on the skin irritation response of guinea pigs, and multiple comparisons were performed. When the data were percentages, they were transformed using the arcsine square root function for processing. All statistical analyses were performed in SPSS version 14.0 (SPSS, Inc., Chicago, IL, USA). Data are expressed as means ± standard errors.

## 3. Results

### 3.1. Alkaloids in Different Body Parts of S. invicta

Eight alkaloids were obtained from different body parts of *S. invicta* workers, and the retention times were between 23 and 36 min (see Figure 1 for the total ion chromatogram of the extract and Appendix A for more detailed mass spectral data). Moreover, five cuticular hydrocarbons were identified, with retention times between 38 and 42 min. The alkaloids extracted from the abdomen (eight alkaloids) included those extracted from the head and thorax (the same four alkaloids) (Figure 1A). The same alkaloids in the body sections were 2-methyl-6-(4′-tridecenyl)piperidine (labeled 2), 2-methyl-6-tridecylpiperidine (labeled 3), 2-methyl-6-(6′-pentadecenyl)piperidine (labeled 5), and 2-methyl-6-pentadecylpiperidine (labeled 7), and their abundances were relatively high (Figure 1B).

The relative alkaloid contents in different parts of *S. invicta* and the total extract obtained from the parts are shown in Figure 2. For sampling location LW, the alkaloids in the abdomen accounted for the highest proportion of total alkaloids (90.25%), followed by those in the thorax (64.39%) and the head (22.05%) (Figure 2A; χ^2^ = 9.74, *df* = 2, *p* = 0.0077), and this result was similar to that of the other two sampling locations (Figure 2B, YW: abdomen 91.50%, thorax 46.88%, head 1.87%; Figure 2C, LD: abdomen 98.16%, thorax 43.69%, head 2.15%). Although there were no quantitative data, we could see from the total ion chromatogram of alkaloids that the cuticular hydrocarbons in the head were the most abundant, followed by those in the thorax (Figure 2D–F).

### 3.2. Effects of Different Extraction Methods on the Purity of Alkaloids

Among the different extraction methods, the abdominal extirpation and venom gland dissection methods yielded alkaloids with the highest purity, with no significant difference between these two methods. The whole-ant solvent-soaking extraction method exhibited the lowest alkaloid purity (Figure 3, *F*_2,17_ = 103.68, *p* < 0.001). The location of the *S. invicta* populations had no effect on the abdominal extirpation and venom gland dissection methods but had a significant effect on the whole-ant soaking method (Figure 3, *F*_1,17_ = 38.79, *p* < 0.05).

### 3.3. Skin Sensitivity Assessment and Histopathological Diagnosis in Guinea Pigs

Erythema and edema appeared in the treated skin patches of guinea pigs 6 h after the AI and BI treatments, respectively, which were very similar to the symptoms of people bitten by *S. invicta* (see Appendix A). After continuous treatment with AI and BI for three days, the skin injury or irritation score reached the maximum and was maintained for 3–4 days afterward, which was significantly different from the VI and SI treatments (day 3 to day 6; Figure 4, *H* = 37.09, *df* =15, *p* = 0.0012). Except for a specific pair of investigation time points (6 h vs. 24 h), there was a significant difference between the AI and BI treatments and the VI and SI treatments (Figure 4, *H* = 142.62, *df* =51, *p* < 0.001). On day 7, scabs appeared in the AI and BI groups, and the symptoms of erythema and edema began to subside slowly; the symptoms subsided significantly faster after the BI treatment (Figure 4, *H* =56.42, *df* = 25, *p* < 0.001). The VI and SI treatments resulted in small red spots and no obvious edema after injection for three consecutive days (see Appendix A), and the above symptoms disappeared after the injection, which showed that the solvent had no effect on the skin of guinea pigs. The change in the skin irritation response score indicated that the AI and BI treatments resulted in similar levels of skin irritation, i.e., they both induced strong sensitization.

On the 4th and 8th days in the AI and BI groups, epidermal cell necrosis at the site of skin tissue injury, nuclear shrinkage, fragmentation, dissolution, and bilateral spinous layer thickening were observed. The collagen fibers of the dermis and subcutaneous tissues and the myocytes of the hypodermis showed large areas of necrosis, accompanied by numerous lymphocytes and neutrophilic infiltration (Figure 5). On the 4th day in the AI and BI groups, compared with the 8th day, small areas of collagen fibers were reduced at the injury site, and the arrangement was sparse and irregular. The skin structures in the VI and SI groups were clear, and no obvious pathological changes were observed (Figure 5).

## 4. Discussion

Continuous stings and attacks by *S. invicta* can lead to serious human diseases, e.g., hemolytic uremic syndrome [31], rhabdomyolysis and acute renal failure [32], and platelet thrombus formation and endothelial injury [14,33], which can lead to death [4]. Moreover, people who are not allergic to the stings also experience conditioned reactions, such as fear and panic, after being stung multiple times by *S. invicta*. The venom released from the venom sacs of IFAs is the cause of these reactions; moreover, *S. invicta* does not lose its stinger upon stinging and may therefore sting an individual multiple times [34]. It is generally believed that the local skin reactions caused by *S. invicta* stings rarely cause serious health problems [21], so research on the inflammation caused by its alkaloids may be ignored. Previous studies found that the piperidine alkaloids in the venom directly caused pronounced dermal necrosis at the site of the stings [31,34]. Additionally, the inhibition of neuronal nitric oxide synthase by venom alkaloids may predispose individuals to experience neurotoxicity and hypercoagulability after extremely large numbers of stings [35]. Although treatment schemes for most insect bites, such as cold compresses or oral anti-allergic drugs, can be applied, an in-depth study on the toxicity of *S. invicta* alkaloids is lacking, which restricts the research and development of targeted treatment schemes. The purpose of the present study was to clarify the distribution of alkaloids in *S. invicta* and then explore a suitable extraction method for obtaining high-purity alkaloids. Finally, we verified the sensitization induced by the extracted alkaloids with an animal model to establish an important scientific basis for the subsequent development of anti-inflammatory drugs for *S. invicta* stings.

Ants have many glands, and these glands secrete various compounds, including alkaloids, which are utilized for defense, attack, individual communication, and disinfection of the nest by workers; these compounds even play an antibacterial role on the egg surface [16,36,37,38]. Ant workers usually disperse venom from the stinger onto the whole body during self-grooming and allogrooming [37,39]; thus, different body parts of *S. invicta* may be contaminated with alkaloids. However, no studies have investigated the difference in or relationship between the presence of alkaloids in different body parts of *S. invicta*. In the present study, we found alkaloids in the head and thorax of *S. invicta* workers but fewer types of alkaloids than in the abdomen. In other words, the alkaloids in the abdomen (eight types) included four types found in other body parts (the head and thorax), although these four alkaloids were present at greater concentrations in the abdomen. Therefore, our results also supported the view of some researchers that the chemical components of alkaloids in *S. invicta* venom can be completely reflected by the whole-ant soaking method [25]. Moreover, we found that the extract from the abdomen contained the highest alkaloid content, followed by that of the thorax and then that of the head. This may be because the venom of ants that sting is primarily composed of alkaloids, and the main structure of the abdomen is the venom reservoir [40]. The postpharyngeal gland of the *S. invicta* head contained most alkaloid types, and its main chemical components [41] were consistent with the alkanes in the body surface compounds of *S. invicta* detected by Nelson et al. [42]. We determined that the main extracts from the head of *S. invicta* were primarily alkanes, with few alkaloids. Four alkaloid types were extracted from all three parts of *S. invicta*, and their proportions out of the total alkaloids in their respective parts were also similar. Thus, the main alkaloids in various body parts were similar, which may be related to the mechanism of enhancing the toxicity of defense products, but this requires further research.

There are many factors affecting *S. invicta* alkaloids, such as individual social form or caste, worker age, geographic location, and season. Deslippe and Guo [43] demonstrated that the ratios of saturated to unsaturated C13 and C15 alkaloids were positively correlated with worker size and increased with worker age. Lai et al. [26] found that the proportions of unsaturated alkaloids differed significantly between the monogyne and polygyne forms of *S. invicta* regardless of growth temperature, sampling season, or geographic location. Cis-C11 was the most abundant alkaloid in queens, followed by trans-C11, while some long-chain alkaloids in workers were not found in queens (all trans structures) [44]. Therefore, we uniformly selected medium-sized workers from polygyne *S. invicta* colonies to avoid these factors affecting alkaloids. However, our study revealed that geographic location also had an effect on the content or proportion of *S. invicta* alkaloids in the same sampling season. We speculate that the reason may be climate differences, different invasion histories, or geographical population differences that existed before the invasion.

Although geographic location impacts the alkaloid content, it does not affect the comparison of high-purity alkaloid extraction methods. We found that the purity of alkaloids extracted by the whole-ant soaking method was the lowest (<90%), while that extracted by the abdominal extirpation method was higher than 97%; there was no significant difference between the methods of abdominal extirpation and venom gland dissection. However, venom gland dissection is more difficult and time consuming than abdominal extirpation. Overall, solvent soaking of whole ants was the simplest method but also showed reduced purity. In terms of subcutaneous immunotherapy, the whole-body extract of *S. invicta* has been widely used because it contains sufficient venom proteins [21,24], and the safety of this method has also been characterized [12]. In a previous study of the effects of *S. invicta* alkaloids on platelet, neutrophil, and histamine release, the extraction or purification of alkaloids was involved; for example, whole ants were exhaustively extracted with methylene chloride in a Soxhlet apparatus [14], or full venom reservoirs were collected and then homogenized in a buffered salt solution [45]. The method of extracting abdomens with venom reservoirs was also mentioned [45], but the purity and applicability of the extraction method have not been clearly described. Although Liu et al. [25] suggested that whole-body solvent soaking adequately extracts venom alkaloids from IFA workers and that venom alkaloids can be separated further by silica gel column chromatography, the alkaloid purity was not high, it cannot simulate the state of alkaloids released by *S. invicta* stings well, and some target substances may be adsorbed in the process of silica gel column chromatography. In other words, alkaloid extraction methods may not be suitable for different experimental purposes.

High-purity venom alkaloids were obtained by abdominal extirpation, and we further verified the similarities in symptoms presented between intradermal injection of alkaloids and *S. invicta* stings through skin irritation tests on guinea pigs, a species that is widely used in skin sensitization tests of chemicals. The skin injury or irritation score is often applied to evaluate skin irritation [30,46,47], and irritation scores between 0 and 4 are used to grade stimulus intensity, which ranges from no response to a severe response. We applied this scoring system and found that the change in the skin irritation response to AI and BI treatments was consistent, i.e., erythema appeared at the beginning, severe abscess and erythema appeared after 3–4 days, and scabbing began after 7 days. The VI treatment also proved that the vehicle added to the alkaloid did not irritate the skin, similar to the control. The skin irritation test not only proved the similarities between alkaloid injection and *S. invicta* stings but also showed that the abdominal extirpation method was effective for extracting high-purity alkaloids. In the past, researchers have investigated *S. invicta* stings with dogs, but the results showed that by 24 h and continuing to the end of the study at 72 h, the sites typically appeared completely normal, and no evidence of anaphylaxis was observed [48]. However, we suggest that these changes in skin inflammation in guinea pigs are similar to those described in humans stung by *S. invicta* according to the skin irritation response and histological changes. There have been many studies on *S. invicta* venom [49,50,51,52], both on venom proteins and alkaloids, but our study confirms the purity of alkaloids extracted with the abdomen extirpation method and the feasibility of simulating *S. invicta* stings in animal skin tests.

## 5. Conclusions

In this study, we first determined the types and contents of alkaloids in various body parts of *S. invicta* to determine the feasibility of using abdominal extirpation to extract alkaloids. After comparing different methods for extracting venom alkaloids, we suggest that abdominal extirpation and venom gland dissection are effective for extracting high-purity alkaloids. Finally, we further proved that alkaloid injection symptoms resemble the symptoms of *S. invicta* stings; we also showed that the abdominal extirpation method is effective for extracting high-purity alkaloids according to the results of skin irritation tests and histopathological diagnosis in guinea pigs. These findings establish a foundation for the development of drugs to treat inflammation from *S. invicta* stings and provide important medicinal insight into the activity and function of *S. invicta* venom alkaloids.

## Figures and Tables

**Figure 1 ijerph-20-01904-f001:**
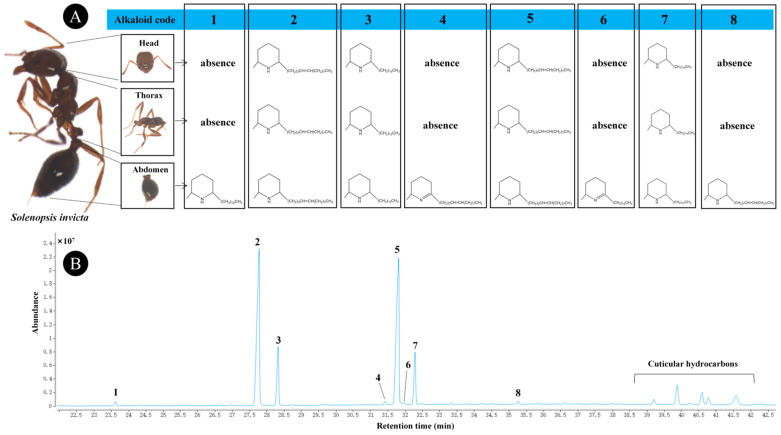
Gas chromatographs of hexane extracts obtained from different body parts (i.e., the head, thorax, and abdomen) of *Solenopsis invicta* workers. The types of alkaloids in these parts of *S. invicta* are shown in (**A**). The names of the various alkaloids (labeled 1–8) are 2-methyl-6-undecylpiperidine, 2-methyl-6-(4′-tridecenyl)piperidine, 2-methyl-6-tridecylpiperidine (3), 2-methyl-6-(6′-pentadecenyl)-2,3,4,5-tetrahydropyridine, 2-methyl-6-(6′-pentadecenyl)piperidine (5), 2-methyl-6-pentadecyl-2,3,4,5-tetrahydropyridine, 2-methyl-6-pentadecylpiperidine, and 2-methyl-6-(8′-heptadecenyl)piperidine, respectively (**B**).

**Figure 2 ijerph-20-01904-f002:**
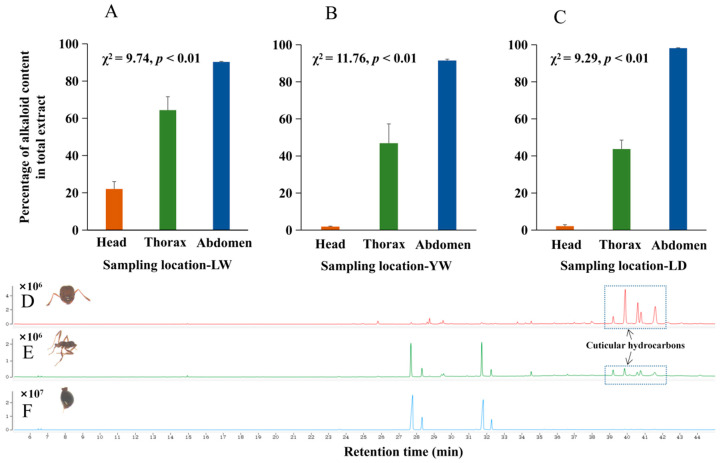
The alkaloid contents in different body parts of the *S. invicta* body relative to that in the total extract obtained from parts. Alkaloid content ((**A**), sampling location was Longwan; (**B**), sampling location was Yiwu; (**C**), sampling location was Liandu) and alkaloid ion chromatogram of the head ((**D**), red line), thorax ((**E**), green line) and abdomen ((**F**), blue line) part of *S. invicta* body. All multiple comparisons were performed using Chi-square tests. Values are the means (+SEs) of triplicate experiments.

**Figure 3 ijerph-20-01904-f003:**
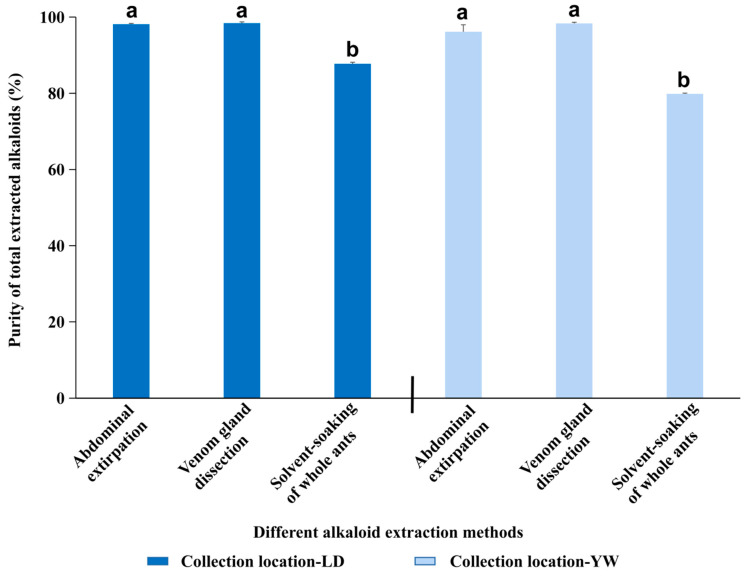
Purity of alkaloids from *S. invicta* obtained by different extraction methods (abdominal extirpation, venom gland dissection, and whole-body solvent soaking) and from different collection locations. Bars labeled with different lowercase letters are significantly different (*p* < 0.05) from each other (different extraction methods) in the same series; all multiple comparisons were performed using Tukey’s HSD test. Values are the means (+SEs) of triplicate experiments.

**Figure 4 ijerph-20-01904-f004:**
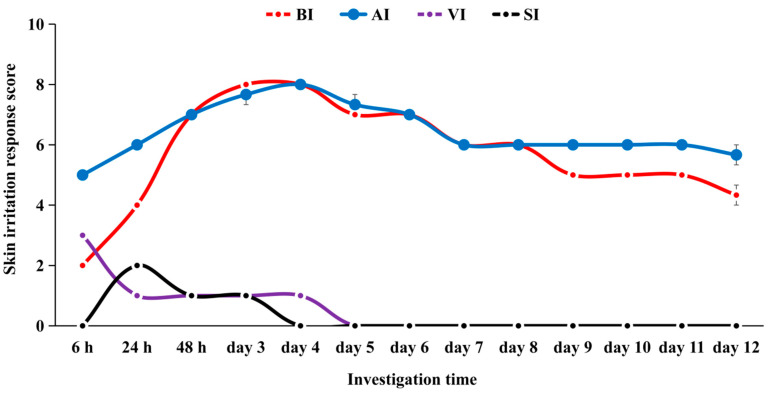
Skin irritation response scores of guinea pigs to intradermal injection treatments at different time points. The four treatment groups were as follows: (1) BI: *S. invicta* stings and bites; (2) AI: Intradermal injection of alkaloids; (3) VI: Vehicle control injection; and (4) SI: Injection of normal saline.

**Figure 5 ijerph-20-01904-f005:**
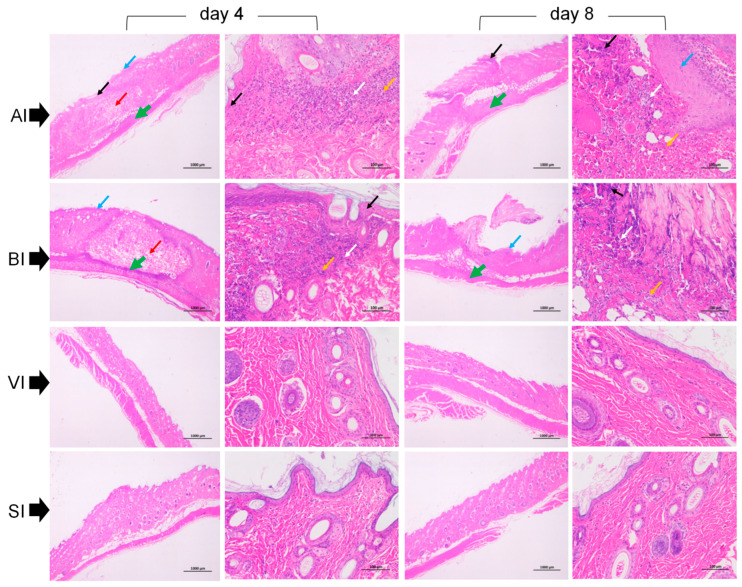
Histological changes in the skin of guinea pigs associated with *S. invicta* stings (**BI**), intradermal injection of alkaloids (**AI**), vehicle control injection (**VI**), and injection of normal saline (**SI**). The left and right vertical images were collected 4 and 8 days after treatment, respectively. The black arrows indicate dissolved epidermal cells or nuclear pyroptosis, the blue arrows indicate thickening of the spinous layer on either side of the epidermal cells, the white arrows indicate dissolved collagen fibers, the orange arrows indicate neutrophilic infiltration of the dermis and subcutaneous tissue, the green arrows (bold line) indicate neutrophilic infiltration of the hypodermis, and the red arrows indicate irregularly arranged collagen fibers in the hypodermis.

## Data Availability

The original contributions presented in the study are included in the article/Appendix A, further inquiries can be directed to the corresponding authors.

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
