# Peer review of "Sensitization of Guinea Pig Skin to Imported Fire Ant Alkaloids and Establishment of an Inflammatory Model"

_ijerph, 2023, doi:10.3390/ijerph20031904_

Round 1

Reviewer 1 Report

This paper describes the effective extraction method for alkaloid components from fire ant Solenopsis invicta, and effects of alkaloids for skin sensitivity in guinea pig. The results are interesting and would be useful for the future development of anti-inflammatory agents for treatment of this ant sting envenomation.

1Although experiments appear to be well performed, the manuscript are not well-prepared. The language of the manuscript should be polished by a native speaker. Prior to acceptance of this work, extensive revisions are needed.

2The venom gland has been shown to be the source of venom, which is found in the abdomen. Why did your research find alkaloids in the heads and chests of red fire ants?

3Generally, mice, rats and New Zealand white rabbits are used as model animals in animal experiments. Why did you choose guinea pigs for your study?

4The applications of alkaloids and ant bitings were repeated in consecutive three days.  How do you determine the effecting time of injected alkaloids? Maybe one-time injection is enough to trigger significant response.

5、The color difference between the yellow and orange arrows in Figure 5 is not significant. It is recommended to change to green or a color that is clearly different from the other colors.

Other minor changes need to be made

Line 91 IFA colonies were, 2 weeks after collecting

Line 94 in different body parts of

Line 95 among body parts

Line 96 change “the species of alkaloids” into “the types of alkaloids and the”

Line 104 change “chest” into “thorax”

Line 207 with the retention times later than 38 min

Lines 208 and 209 change “released from” into “extracted from”

Line 208 the same four alkaloids

Line 216 different body parts

Line 217 the types of alkaloids

Author Response

Comment 1: Although experiments appear to be well performed, the manuscript are not well-prepared. The language of the manuscript should be polished by a native speaker. Prior to acceptance of this work, extensive revisions are needed.

Our response: We have polished the manuscript for language and grammar by a native speaker from the team of American Journal Experts.

Comment 2: The venom gland has been shown to be the source of venom, which is found in the abdomen. Why did your research find alkaloids in the heads and chests of red fire ants?

Our response: According to the current references, the alkaloids in the head and thorax of red fire ant may come from, 1) Workers disperse venom from sting and to whole body via selfgrooming and allogrooming. 2) When the queens or workers use alkaloids to disinfect the micro-environment or egg surface, the workers epidermis is contaminated with alkaloids. So alkaloids were also detected in different parts of the individual ants that we sampled.

Related references:

Ref. 1. Antibacterial properties of contact defensive secretions in neotropical Crematogaster ants. (DOI: 10.1590/S1678-91992012000400013).

Ref. 2. Colonization by the Red Imported Fire Ant, Solenopsis invicta, Modifies Soil Bacterial Communities. (DOI: 10.1007/S00248-021-01826-4).

Ref. 3.Gaster flagging by fire ants (Solenopsis spp.): Functional significance of venom dispersal behavior. (DOI: 10.1007/bf01012125).

Comment 3: Generally, mice, rats and New Zealand white rabbits are used as model animals in animal experiments. Why did you choose guinea pigs for your study?

Our response: Because the skin of guinea pigs is very sensitive, compared with ICR mice and New Zealand white rabbits, guinea pigs are more similar to human skin, so most of the current research uses guinea pigs for skin model testing. In addition, we have done preliminary experiments with ICR mice, New Zealand white rabbits and guinea pigs, and found that the effect of biting guinea pigs is similar to that of humans.

Comment 4: The applications of alkaloids and ant bitings were repeated in consecutive three days. How do you determine the effecting time of injected alkaloids? Maybe one-time injection is enough to trigger significant response.

Our response: In the preliminary experiment, we found that the guinea pig's reaction was small on the first day, increased on the second day, and was most obvious on the third day, which is similar to the symptoms of human bites. Therefore, in order to make the symptoms last longer, we use three consecutive days of injection for maximum response.

Comment 5: The color difference between the yellow and orange arrows in Figure 5 is not significant. It is recommended to change to green or a color that is clearly different from the other colors.

Our response: According to the reviewer's suggestion, the yellow arrow has been changed to the green arrow; At the same time, the lines of the green arrows were made thicker to account for the red-green color blind readers. We have uploaded the latest modified figure.

Other minor changes also been made.

Reviewer 2 Report

Invasive insect, imported fire ants (IFAs), could induce inflammation, allergies, shock and even death. It is widely believed that IFA venom protein could be responsible for its pathogenesis of IFA sting induced. In this report, the authors focus on the potential toxicity or pathogenicity of alkaloids, the main substances in IFA venom. They have provided several evidences to suggest it is important for the alkaloids to induce toxicity in a model animal guinea pigs. I would like to suggest they could provide more quantitative data about alkaloids concentration in each of body part and also in the solvent-soaking extraction in their animal model experiment. 

Author Response

Comment 1: I would like to suggest they could provide more quantitative data about alkaloids concentration in each of body part and also in the solvent-soaking extraction in their animal model experiment.

Our response: Thank you for your advice. According to the animal experiments in this paper, we believe that the sensitization reaction of red fire ant to guinea pig is mainly due to its stinging and contact with alkaloids in the abdomen, while the proportion of contact with alkaloids in other body parts is very low. However, if future animal experiments involve the study on the sensitization difference of alkaloids with different concentrations, such as the alkaloids in the palate (bite) and the abdomen (sting), we think that more quantitative data about alkaloids concentration from different body parts can be provided. Your proposal could provide new ideas for broadening the research. As for the alkaloids concentration used in animal model test, we have added relevant explanations to the text (line 164-166).

Reviewer 3 Report

See attach.

Author Response

Comment 1: The sentence ‘histopathologic diagnosis suggest that the extracted alkaloids produced symptoms effectively simulating those of IFA stings by intradermal injection’ could be better expressed as ‘histopathologic diagnosis suggest that the intradermal injection of the extracted alkaloids produced symptoms effectively simulating those of IFA stings.’

Our response: Done.

Comment 2: Insert ‘and’ to say ‘alkaloids, and proportions’.

Our response: Done.

Comment 3: Were the ‘stationary workers’ maintained at room temperature and therefore unfrozen? If this is so, I suggest to change ‘stationary’ by ‘unfrozen’, ‘remaining’ or alike.

Our response: “stationary” has been replaced with “remaining”.

Comment 4: ‘vitamin C’ is repeated.

Our response: Deleted.

Comment 5: It is shorter to say ‘standard laboratory conditions: temperature 22-25 °C, humidity 50-60%, and 12 h/12 h light/dark cycle’.

Our response: Done.

Comment 6: If I am not wrong, the total volume of the solution for the alkaloids apparently amounts to 3.432 mL (1560 µL of 1% HCl +1560 µL of 0.9% NaCl + 312 µL of 1% NaOH), not 3.9 mL as cited (?). Please, check.

Our response: Thank you for your care. Your calculation was correct, our sentence expression was not clear enough, the solution was finally fixed to 3.9 mL. We've changed the statement.

Comment 7: It is said ‘0.05 mL of alkaloids were injected into the exposed patch of skin on the guinea pigs’(?), or were 0.05 mL of the alkaloids solution so that the amount of alkaloids per injection was 0.05 mL x 4.0 mg/mL = 0.20 mg of pure alkaloids?

Our response: Yes, we mean 0.05 mL of the alkaloids solution, containing 0.20 mg of pure alkaloids. We've changed the statement.

Comment 8: In the irritation scores, grades 2 and 3 indicate ‘slight edema’ in both cases. Please, revise.

Our response: Done. We've revised the statement.

Comment 9: L185-8. This sentence is too long, transform it into two.

Our response: Done.

Comment 10: Figure 1. Lettering of the alkaloids formula is very small and almost unreadable. All alkaloids should be numbered at least in text.

Our response: We've replaced it with a clear figure. Eight alkaloids have been numbered (labeled from 1 to 8) in text and figure.

Comment 11: Give the retention time range of the cuticular hydrocarbons and delete ‘that was 38 min later’.

Our response: Done.

Comment 12: Figure 2 legend. It should contain the different sampling locations in A, B and C.

Our response: Done.

Comment 13: Introduce comma after ‘reactions’ and ‘panic’.

Our response: Done.

Comment 14: L318-20 and below. ‘Alkaloid species’(?). This is misleading since species preferentially refer to animals (ants in this case) not to chemicals. I suggest to say, f.i., ‘alkaloid types’. References.

Our response: “species” has been replaced with “types” as suggested in the whole text.

The references section has been revised as suggested.

Round 2

Reviewer 2 Report

The second version has addressed my concerns. No other comment here.